# The female protective effect against autism spectrum disorder

## Graphical abstract

## Authors

Emilie M. Wigdor, Daniel J. Weiner, Jakob Grove, ..., Somer L. Bishop, Anders D. Børglum, Elise B. Robinson

## Correspondence

erob@broadinstitute.org

## In brief

Wigdor et al. find evidence supporting a female protective effect against autism spectrum disorder (ASD): (1) siblings of female ASD probands are more likely to be diagnosed with ASD than siblings of male ASD probands and (2) mothers carry more common, inherited genetic risk for ASD than fathers. Taken together, these results emphasize the breadth of the role of sex in ASD risk and could impact the design and interpretation of genetic and neurobiological studies of ASD.

## Highlights

- Evidence of female protective effect against ASD from common, inherited variation

- Evidence of FPE in both affected and unaffected members of ASD-impacted families

- Mothers of children with ASD carry more genetic risk for ASDs than fathers

Wigdor et al., 2022, Cell Genomics 2, 100134
June 8, 2022 © 2022 The Authors.

# Cell Genomics

CellPress

## Article

# The female protective effect against autism spectrum disorder

Emilie M. Wigdor,[1,2] Daniel J. Weiner,[1,3] Jakob Grove,[4,5,6,7] Jack M. Fu,[1,8] Wesley K. Thompson,[9] Caitlin E. Carey,[1,3] Nikolas Baya,[1,3] Celia van der Merwe,[1,3] Raymond K. Walters,[1,3] F. Kyle Satterstrom,[1,3] Duncan S. Palmer,[1,3] Anders Rosengren,[7,10] Jonas Bybjerg-Grauholm,[7,16] iPSYCH Consortium,[17] David M. Hougaard,[7,16] Preben Bo Mortensen,[4,10,12,13] Mark J. Daly,[1,3,11] Michael E. Talkowski,[1,8] Stephan J. Sanders,[14] Somer L. Bishop,[14] Anders D. Børglum,[4,5,10] and Elise B. Robinson[1,3,15,18,*]

[1]Stanley Center for Psychiatric Research, Broad Institute of MIT and Harvard, Cambridge, MA 02142, USA
[2]Wellcome Trust Sanger Institute, Hinxton CB10 1SA, UK
[3]Analytic and Translational Genetics Unit, Department of Medicine, Massachusetts General Hospital, Boston, MA 02114, USA
[4]Center for Genomics and Personalized Medicine (CGPM), Aarhus University, 8000 Aarhus, Denmark
[5]Department of Biomedicine (Human Genetics) and iSEQ Center, Aarhus University, 8000 Aarhus, Denmark
[6]Bioinformatics Research Centre, Aarhus University, 8000 Aarhus, Denmark
[7]The Lundbeck Foundation Initiative for Integrative Psychiatric Research, iPSYCH, 8210 Aarhus, Denmark
[8]Center for Genomic Medicine, Massachusetts General Hospital, Boston, MA 02114, USA
[9]Laureate Institute for Brain Research, Tulsa, OK 74136, USA
[10]Institute of Biological Psychiatry, MHC Sct Hans, Copenhagen University Hospital, 4000 Roskilde, Denmark
[11]Finnish Institute for Molecular Medicine, University of Helsinki, 00290 Helsinki, Finland
[12]National Center for Register-Based Research, Aarhus University, 8210 Aarhus, Denmark
[13]Center for Integrated Register-based Research, Aarhus University, 8210 Aarhus, Denmark
[14]Department of Psychiatry and Behavioral Sciences, UCSF Weill Institute for Neurosciences, University of California, San Francisco, San Francisco, CA 94158, USA
[15]Department of Epidemiology, Harvard T.H. Chan School of Public Health, Boston, MA 02115, USA
[16]Center for Neonatal Screening, Department for Congenital Disorders, Statens Serum Institut, 2300 Copenhagen, Denmark
[17]A list of members and affiliations appears at the end of the paper
[18]Lead contact
*Correspondence: erob@broadinstitute.org

## SUMMARY

Autism spectrum disorder (ASD) is diagnosed three to four times more frequently in males than in females. Genetic studies of rare variants support a female protective effect (FPE) against ASD. However, sex differences in common inherited genetic risk for ASD are less studied, particularly within families. Leveraging the Danish iPSYCH resource, we found siblings of female ASD cases (n = 1,707) had higher rates of ASD than siblings of male ASD cases (n = 6,270; p < 1.0 × 10⁻¹⁰). In the Simons Simplex and SPARK collections, mothers of ASD cases (n = 7,436) carried more polygenic risk for ASD than fathers of ASD cases (n = 5,926; 0.08 polygenic risk score [PRS] SD; p = 7.0 × 10⁻⁷). Further, male unaffected siblings under-inherited polygenic risk (n = 1,519; p = 0.03). Using both epidemiologic and genetic approaches, our findings strongly support an FPE against ASD's common inherited influences.

## INTRODUCTION

Autism spectrum disorder (ASD) is diagnosed three to four times more frequently in males than in females.[1–3] The possibility of a "female protective effect" (FPE) against ASD has been described extensively and has received consistent support from the results of genetic studies of *de novo* variants.[4–13] Many types of ASD-associated *de novo* variants are observed more frequently in female cases.[4–13] In general, the more ASD risk carried by a *de novo* variant class, the greater its overrepresentation among affected females.[8] This suggests that, on average, females accumulate more risk than males before being ascertained as ASD cases.

Male-female differences are less clear in the context of ASD's common, inherited genetic influences, which constitute the majority of genetic risk for ASD.[14] Given the findings above, we may expect elevated polygenic risk for ASD in female cases; however, that has not been consistently observed.[4,15,16] Inconsistent observations could be a function of statistical power, as the polygenic risk score (PRS) for ASD currently explains limited case-control variance on the liability scale (<3%), and under 4,000 female cases are present in published ASD genome-wide association study (GWAS) meta-analyses.[4,15] A recent study found evidence for increased burden of combination polygenic risk (ASD + schizophrenia + educational attainment) in female

ASD cases,[16] further suggesting a male-female difference may appear using the ASD PRS alone were it better powered.

In this study, we used two complementary strategies to better understand the relationship between sex and inherited genetic risk for ASD. We first conducted a large sibling recurrence analysis, leveraging the Danish Lundbeck Foundation Initiative for Integrative Psychiatric Research (iPSYCH) resource. We then examined the relationship between sex and common, autosomal polygenic risk for ASD in whole families, focusing on both affected and unaffected family members.

Under the FPE model, one expects a greater aggregation of ASD risk in female cases than in male cases. In the context of inherited genetic risk, which is shared within families, that expectation extends to the family members of female cases. For example, we expect siblings of female ASD cases to carry more risk for ASD than siblings of male ASD cases, regardless of whether they are categorically affected themselves.[17] Sibling recurrence is a particularly useful metric of inherited or familial risk. Full siblings share 50% of their segregating DNA variants and are typically close enough in age to share diagnostic environments. Shared diagnostic environment is important when considering ASD recurrence. The estimated prevalence of ASD has increased over 30-fold over the last four decades,[18] primarily due to diagnostic expansion.[19,20] Members of previous generations, particularly those able to live independently as adults, were far less likely to receive an ASD diagnosis in childhood than children born as of writing.[19,20] For this reason, inclusion of parents or aunts and uncles in familial recurrence analyses can complicate data interpretation. Our analysis was accordingly limited to siblings.

Several previous studies have considered the FPE through familial recurrence, with inconsistent results.[21–24] To improve data interpretability, we used national patient registry data and stratified ASD cases based on presence or absence of co-diagnosed intellectual disability (ID). Despite sharing the majority of their rare variant influences,[7] ID and ASD do not appear to share their common polygenic influences: as currently estimated, the genetic correlation between ID and ASD is not significantly different from zero.[25] Further, evidence suggests reduced SNP heritability for forms of ASD in which co-diagnosed ID is more common.[15,25] As (1) lower heritability predicts lower familial recurrence and (2) ascertained female ASD cases are more likely to have co-diagnosed ID, failing to stratify by ID could render a male-female comparison difficult to interpret. Our recurrence analyses focused on ASD without co-diagnosed ID (from here: *ASDnoID*) and used ID without co-diagnosed ASD (from here: *IDnoASD*) as a negative control. We excluded individuals with diagnoses of both ASD and ID (approximately 15% of ASD cases in Denmark), as there were too few cases in that group for an independent sibling recurrence analysis (n = 372 female cases with at least one sibling). We then complement the epidemiologic analyses with a statistical genetic comparison using multiple members of ASD-affected families and a new ASD PRS from a large, unpublished GWAS meta-analysis.

## RESULTS

### FPE and sibling recurrence

The Danish Psychiatric Central Research Register and the Danish National Patient Register are unique resources, well suited to careful consideration of sibling recurrence. They are complete until 2012 and 2013, respectively, and contain medical record data on the entire Danish population born between May 1, 1981 and December 31, 2005 (n = 1,472,762). We linked the psychiatric and patient registers to find all Danish families with two or more full siblings born during this time period. We identified 94,790 such families. We then identified the families with at least one child with *ASDnoID* or *IDnoASD*. This analysis included all diagnosed *ASDnoID* and *IDnoASD* cases in this population during this period. When a family included more than one affected child, we selected one at random to be the "index case" (from here: cases). We analyzed one sibling per family; if the family included more than one sibling, we selected one at random for inclusion in the analysis. We examined ASD and ID diagnoses in the selected siblings. As the focus of the analysis was recurrence of ASD and ID and any selection among siblings was performed at random, sibling selection was not diagnosis dependent (i.e., if the family included a sibling with ASD and a sibling without, either could be selected, with equal probability). A detailed description of this process can be found in the STAR Methods: Sibling recurrence of ASD and ID.

To investigate the FPE, we examined whether siblings of female cases of *ASDnoID* (n = 1,707 siblings) have higher risk for ASD and/or ID themselves than the siblings of male cases of *ASDnoID* (n = 6,270 siblings). We were adequately powered to examine co-occurring ASD and ID (*ASDandID*) as an outcome in the siblings. In siblings, there were accordingly three potential outcomes: *ASDnoID*, *ASDandID*, and *IDnoASD*. We estimated sibling risk by comparing diagnosis rates in the siblings with diagnosis rates in age- and sex-matched controls, drawn at random from the Danish population. To increase power, we used 2:1 control to case matching. We followed the same procedures for siblings of female cases of *IDnoASD* (n = 506 siblings) and siblings of male cases of *IDnoASD* (n = 811 siblings).

The primary results are presented in Figure 1. An odds ratio (OR) of more than 1 suggests that case siblings were more likely to receive a diagnosis than age- and sex-matched individuals from the general population. Siblings of female *ASDnoID* cases were approximately seven times as likely (OR = 7.19; 95% confidence interval [CI] = 5.09–10.09) to receive a diagnosis of *ASDnoID* themselves than a general population individual. For siblings of male *ASDnoID* cases, there was a nearly 4-fold (OR = 3.76; 95% CI = 3.10–4.54) increase in risk. In fact, while all siblings of *ASDnoID* cases were at increased ASD risk (p < $1.34 \times 10^{-4}$ for all comparisons), the siblings of female *ASDnoID* cases were at even greater risk than the siblings of male *ASDnoID* cases (p < 0.01 for both comparisons). This is consistent with expectations of the FPE. We only compared risk between siblings of female and male cases if both sibling groups showed elevated risk against the general population. This is akin to only testing for an interaction in the presence of significant main effects.

The pattern was different for the siblings of *IDnoASD* cases. First, neither siblings of female cases (n = 506; *ASDandID*: OR = 2.00, 95% CI = 0.12–32.07; *ASDnoID*: OR = 2.01, 95% CI = 0.80–5.12) nor siblings of male cases (n = 811; *ASDandID*: OR = 6.02, 95% CI = 0.63–57.95; *ASDnoID*: OR = 1.49, 95% CI = 0.79–2.80) showed increased risk for ASD (with or without

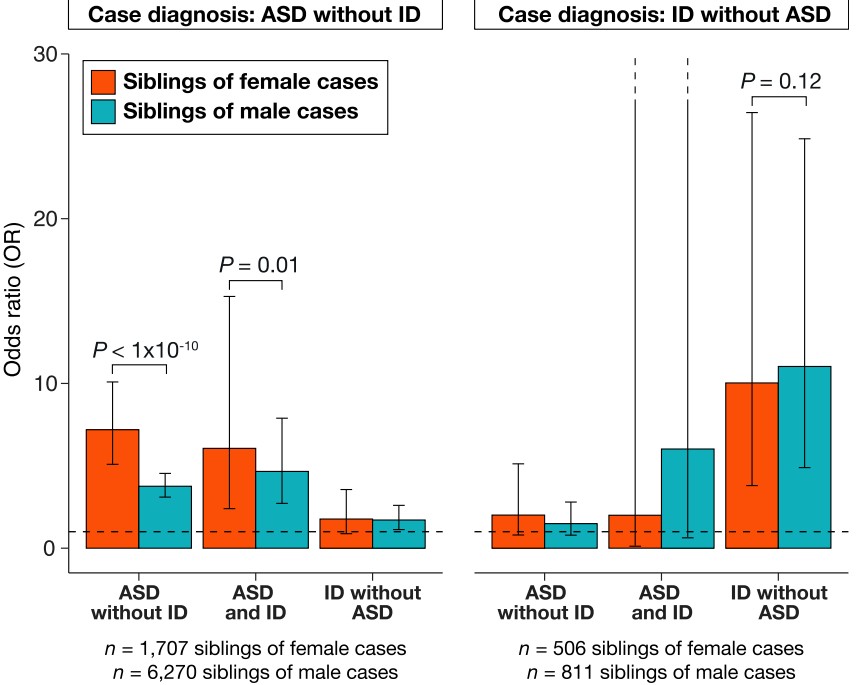

**Figure 1. Sibling recurrence of ASD and ID**
Red bars represent odds ratios (ORs) for siblings of female cases, and teal bars represent ORs for siblings of male cases. ORs indicate the increase in risk for each diagnosis among siblings of cases, as compared with age- and sex-matched controls, derived from logistic regression (STAR Methods; Sibling recurrence of ASD and ID). Error bars represent 95% confidence intervals. p values are from a Wald test to determine whether ORs are significantly different from one another. p values for the male-female comparison were only calculated when both ORs were significantly different from 1. Underlying data are in Tables S1 and S2.

obvious and substantial concentration of ASD-like traits, the family could not participate in the study.[26] Families with ASD-diagnosed parents can participate in SPARK, but we excluded these families from our analysis. SPARK parents remaining in the analysis could still have a substantial aggregation of ASD symptomatology.

We expect mothers and fathers of children with ASD to carry elevated ASD risk relative to the general population. To estimate this increased risk, we integrated the SSC and SPARK data with a large general population cohort, the UK Biobank (UKB).[29] Using standard deviations (SDs) on the UKB ASD PRS distribution as our scale, we then estimated the burden of common polygenic risk for ASD in all European ancestry parents in SPARK and SSC, as well as in ancestry-matched controls from UKB, controlling for the first 15 principal components (PCs) of ancestry. As expected, parents of ASD cases carried more genetic risk for ASD than controls (0.23 SD; $p = 1.9 \times 10^{-7}$; Figure 2).

Under an FPE model, mothers would, on average, be able to carry more ASD risk than fathers before meeting ASD case criteria. Consistent with FPE expectations, we found that mothers of ASD cases carried significantly more polygenic risk for ASD than fathers of ASD cases (n = 7,436 mothers; n = 5,926 fathers; 0.09 SD; $p = 7.0 \times 10^{-7}$; Figure 2). The increase in ASD PRS in ASD mothers compared with females in the general population was about 50% greater than the increase in ASD PRS in ASD fathers compared with males in the general population. This mother-father difference is present independently in both SSC (n = 2,061 mothers; n = 2,079 fathers; 0.08 SD; $p = 8.0 \times 10^{-3}$) and SPARK (n = 5,375 mothers; n = 3,847 fathers; 0.09 SD; $p = 5.2 \times 10^{-5}$). It is also present when comparing full trios: families where both parents are present in the dataset (n = 4,809 complete trios; $p = 1.4 \times 10^{-5}$). Further, while ASD cases had significantly greater PRS for ASD than their unaffected mothers on average (n = 7,628; 0.09 SD; $p = 1.2 \times 10^{-8}$; Figure 2), that elevation was strikingly similar to the elevation observed between mothers and fathers. At this sample size, there is no sex difference in ASD PRS in UKB (p = 0.15). This is expected of any population sample when using an autosomally constructed PRS.

Finally, we compared the polygenic burden of male and female ASD probands, controlling for comorbid ID (STAR

co-diagnosed ID) at these sample sizes. As increased risk for ASD could not be detected, we did not test for a difference in ASD risk between siblings of female versus male *IDnoASD* cases. The siblings of *IDnoASD* cases were, however, at significantly increased risk for *IDnoASD* themselves ($p < 3.13 \times 10^{-6}$ for both comparisons). This was true for both siblings of male cases and the siblings of female cases. Sibling risk of *IDnoASD* recurrence did not significantly differ by the sex of the *IDnoASD* case (p = 0.12).

We were not statistically powered to simultaneously consider sex of the case and sex of the sibling. However, in an analysis of risk to male versus female siblings of all ASD cases, risk did not differ meaningfully by sex of the sibling when using a sex-specific general population rate (Figure S1; Table S8; STAR Methods: Sibling recurrence of ASD and ID; Methods S1: Sibling recurrence of ASD and ID, by sibling sex).

## FPE and ASD parents
We next examined the FPE in two genetically characterized ASD cohorts: the Simons Simplex Collection (SSC)[26] and the Simons Foundation Powering Autism Research for Knowledge (SPARK) cohort.[27,28] The SSC consists of families with one affected child and two confirmed unaffected parents. SPARK includes families with a variety of structures.

Parent-child designs present an opportunity to examine the role of the FPE in parents of cases, as well as in ASD cases themselves. We expect parents of ASD cases to have greater than average risk for ASD, simply because they have a child with ASD. The parents, however, are usually categorically unaffected. Some ASD studies, like the SSC, screened parents for ASD and ASD-like symptomatology. If a parent met criteria for an ASD diagnosis or had an

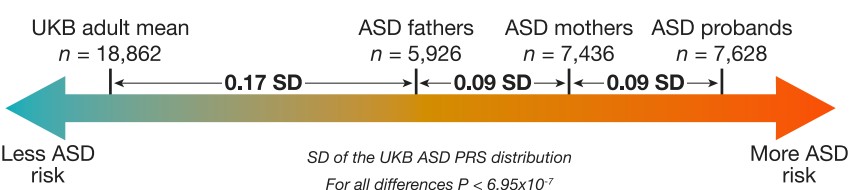

ASD risk continuum

UKB adult mean
*n* = 18,862

ASD fathers
*n* = 5,926

ASD mothers
*n* = 7,436

ASD probands
*n* = 7,628

←——— 0.17 SD ———→ ← 0.09 SD → ← 0.09 SD →

Less ASD
risk

*SD of the UKB ASD PRS distribution*
*For all differences P < 6.95x10⁻⁷*

More ASD
risk

**Figure 2. The continuum of ASD polygenic risk in the general population and families with an ASD case**
Between-group differences in polygenic score for ASD and p values from linear regression comparing group polygenic scores while controlling for 15 principal components of ancestry. ASD groups are combined across the SSC and SPARK collections. Autosomal polygenic risk scores were calculated using weights from a GWAS of ASD cases (n = 19,870) and controls (n = 39,078) from the iPSYCH consortium in Denmark (STAR Methods: Generation of polygenic risk score). Group differences are standardized using the UK Biobank ASD PRS distribution. Underlying data are in Tables S3–S6.

Methods: Polygenic risk comparisons). As a greater fraction of female probands have comorbid ID, ID could otherwise confound this comparison. We thus restricted the analysis to probands with measured IQ and defined ID as full-scale IQ < 70 in SSC or a notation of "cognitive impairment" in SPARK. As expected under a FPE, we observed nominally higher ASD polygenic burden in female compared with male probands (0.08 SD; p = 0.03; n = 789 male probands with ID; n = 230 female probands with ID; n = 3,422 male probands without ID; n = 662 female probands without ID).

### FPE and the polygenic transmission disequilibrium test (pTDT)

The pTDT compares polygenic risk between parents and their children. It leverages the expectation that, in a random sample of parent-child trios, the mean of the children's PRS for any trait will equal the mean of the mid-parent PRS (defined as the average of the mothers' and fathers' PRSs). Ascertainment for a phenotypic deviation between children and parents, for example, sampling children with ASD and parents without ASD, breaks that expectation and allows one to identify polygenic risk factors that are associated with the ascertained outcome. We have previously shown that children with ASD, on average, substantially over-inherit their parents' polygenic risk for ASD, as well as for schizophrenia and increased educational attainment.[4]

Larger ASD datasets, in conjunction with a new and better-powered ASD PRS, allow us to revisit pTDT in light of the differential parental polygenic risk (Figure 2). The difference in average ASD PRS between case mothers and case fathers changes our understanding of the mid-parent PRS. On average, male siblings of children with ASD are now expected to inherit more risk for ASD than is carried by their fathers (Figure 3). To the extent that the mean difference in parental PRS reflects a sex difference in ASD risk tolerance, male siblings have substantially increased risk compared with female siblings. The difference in ASD PRS between ASD case mothers and fathers should be better tolerated in female siblings than in male siblings. The average mid-parent risk is less than the average risk carried by unaffected mothers of ASD cases, meaning females can tolerate higher risk than that expected in female siblings.

To investigate the FPE throughout families affected by ASD, we identified families in SSC and SPARK that include (1) an affected child, (2) two unaffected parents, and (3) an unaffected sibling and performed pTDT on male and female unaffected siblings (n = 1,519 males; n = 1,611 females; STAR Methods:

Polygenic risk comparisons). We found that male unaffected siblings significantly under-inherit their parents' polygenic risk for ASD (p = 0.03; Figure 3). This is consistent with an average requirement for their PRS to decline from the mid-parental PRS to around that of their unaffected fathers, in order to remain unaffected themselves. We did not see a deviation from expectation in female siblings (p = 0.39; Figure 3). While this is consistent with the FPE, the difference in transmission between male and female siblings is not statistically significant and should be re-investigated with larger samples.

We used exome sequence data from SSC and SPARK to identify the subset of ASD cases carrying a high-impact *de novo* variant, specifically predicted to disrupt the function of a constrained gene (12% of cases across both cohorts; see STAR Methods: De novo variant analysis). We hypothesized that high-impact *de novo* variants and the FPE create differences in the amount of liability space remaining to be filled by common polygenic variation. These differences may create the following ordering of polygenic over transmission (lowest to highest): (1) male cases with a high-impact *de novo* variant (n = 436), (2 and 3) either female cases with a high-impact *de novo* variant (n = 159) or male cases without a high-impact *de novo* variant (n = 3,468), and (4) female cases without a high-impact *de novo* variant (n = 757).

The pTDT results reflected this expected gradient (Figure 3). Male probands with high-impact *de novo* variants had the lowest polygenic over-inheritance (0.08 SD; p = 0.10), which was not significantly different from mid-parent expectation and was similar to that of their unaffected mothers (0.06 SD from the mid-parent value). Female cases without a high-impact *de novo* variant had nearly three times the polygenic over-inheritance (0.23 SD; p = 7.82 × 10⁻¹¹) of male cases with a high-impact *de novo* variant (p = 0.02).

### DISCUSSION

Evidence from multiple types of genetic risk, and multiple members of families affected by ASD, supports a FPE model, in which females have a higher liability threshold for receiving a diagnosis of ASD. We note that, in this analysis, female protection and male risk are one and the same. With only two categories and no insight into mechanism, they are in fact indistinguishable. We also note that polygenic risk for ASD is, in the general population, associated with many positive traits.[4,15,30] Dozens of studies have noted a positive, general population correlation between polygenic risk for ASD and greater educational attainment,

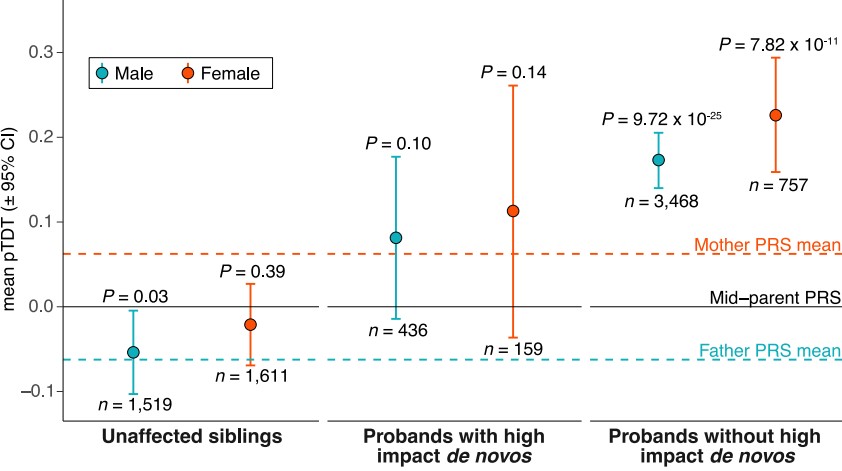

**Figure 3. Polygenic transmission disequilibrium in ASD cases and unaffected siblings**
Transmission disequilibrium standardized to the mid-parent PRS distribution with error bars denoting 95% confidence intervals. p values are from a two-sided, one-sample t test and estimate the probability that polygenic deviation is equal to 0. Cases and controls are combined across SSC and SPARK cohorts. The mother and father PRS mean lines are the mean values from pTDT of each parent against the mid-parent expectation (symmetric by definition). Summary statistics for the PRS are from a GWAS of ASD cases (n = 19,870) and controls (n = 39,078) from the iPSYCH consortium in Denmark (STAR Methods: Danish ASD GWAS). Underlying data are in Table S7.

stronger reasoning ability, and many other beneficial attributes in a cognitively demanding economy. In females, the ability to tolerate more ASD risk without manifesting some of the more isolating elements of diagnosed ASD can benefit individuals, families, and communities. While one may be tempted to quantify a formal expectation of ASD's genetic architecture under specified circumstances (e.g., female with a high-impact *de novo* variant; male without), such expectations would depend on a stable, or at least fairly predictable, phenotype. ASD, as currently diagnosed, is neither. There are predictable elements of sex by phenotype interaction in diagnosed cases, for example, escalating male-to-female ratio with increasing case IQ.[31] However, even after conditioning on IQ, one is left with residual phenotypic associations to sex among ascertained cases. For example, females are on average diagnosed later than males.[20] Similarly, sex differences in genetic architecture remain after conditioning on presence or absence of a strong acting *de novo* variant. Across individuals with ASD, *de novo* variant count is associated with variant impact: as *de novo* variant count increases, so does their average effect size contribution to ASD.[4] Fewer of the variants are benign; more are likely clinically returnable.

Further, one must make several assumptions in order to easily interpret a PRS comparison between male and female cases. For example, one must assume equivalent genetic architecture between ASD as diagnosed in males (male ASD) and as diagnosed in females (female ASD). The previously described differences in rare variant burden, along with preliminary evidence from studies of SNP heritability, already violate that assumption.[5–8,15] In addition, one needs to assume that male ASD and female ASD have equivalent polygenic influences (a genetic correlation of 1). This is unclear at current sample sizes.[15] Even once that analysis becomes adequately powered, the correlation will be difficult to interpret. The male-to-female ratio in ASD increases with increasing case IQ, and this brings with it additional average differences in behavioral, cognitive, and medical comorbidities.[19] Any estimated genetic correlation between male and female ASD could accordingly conflate sex-based and phenotype-based heterogeneity.

We do not know what renders females more tolerant of ASD's genetic risk factors or what, if anything, the mechanisms underlying that tolerance have in common with ASD genetic risk. Analysis at the molecular level will be necessary to address that question. At the statistical level, assuming adequate phenotypic stability and characterization, increasing sample sizes will lead to increasingly clear male-female differences. Future studies can further explore this axis of heterogeneity in ASD.

## Limitations of the study

This study has several limitations. The true ID rate in ASD cases in Denmark is likely higher than reported. If consistent with the rate of ID in ASD cases in the United States or the United Kingdom, it would be approximately 40% over this diagnostic period.[20] ID in the context of ASD is often underreported in medical record and registry data, as it is rarely prescription associated. If comorbid ID was in fact present in "ASD no ID" index cases, we would expect their siblings to be more likely to receive a diagnosis, which would increase overall recurrence rates among siblings and bias our results toward the null hypothesis. We could not attempt to identify additional individuals with ID through information on educational attainment, standardized testing, or assessments of cognitive performance, as these are not linked to the Danish medical registry. We are also limited by the relative scarcity of *IDnoASD* diagnoses in this dataset. A recent nationally comprehensive survey of the Danish registry data noted that, by age 18, the cumulative incidence of ID diagnoses in males (1.5%) is lower than the cumulative incidence of ASD diagnoses in females (1.9%).[32] Our exclusion of case children with both ID and ASD, along with the analytic requirement for two-child families, rendered the *IDnoASD* analyses small in comparison to those focused on ASD alone.

It is worth noting that the influences on differential rates of ASD diagnosis are clearly multifactorial, extending beyond solely genetic influence. One well-known influence is diagnostic bias, which may occur for many reasons, including societal norms of behavior, bias in assessment tools, the sex of evaluators, misdiagnosis of female cases, better "masking" of autistic traits in

females, and sex differences in internal and externalizing features of autism.[3,33]

## STAR★METHODS

Detailed methods are provided in the online version of this paper and include the following:

- KEY RESOURCES TABLE
- RESOURCE AVAILABILITY
  - Lead contact
  - Materials availability
  - Data and code availability
- EXPERIMENTAL MODEL AND SUBJECT DETAILS
  - Simons simplex collection (SSC)
  - Simons foundation powering autism research for knowledge (SPARK)
  - UK Biobank
  - iPSYCH
- METHOD DETAILS
  - Identifying families in Danish registry data
  - Sibling recurrence of ASD and ID
  - Danish genotype data imputation
  - Danish ASD GWAS
  - SSC imputation
  - SPARK imputation
  - *De novo* variant analysis
  - Ancestry definition in SSC, SPARK and UKB
  - Generation of polygenic risk score
  - Polygenic risk comparisons
- QUANTIFICATION AND STATISTICAL ANALYSIS

### SUPPLEMENTAL INFORMATION

### ACKNOWLEDGMENTS

This work was supported by the Autism Science Foundation and Hilibrand Family Foundation (ASP 001 to S.J.S., ASP 002 to S.L.B., and ASP 003 to E.B.R.), the NIMH (RMH111813A to E.B.R., U01MH111662 to S.J.S., and F30MH129009 to D.J.W.), and the NLM (T15LM007092 to D.J.W.). The iP-SYCH team was supported by grants from the Lundbeck Foundation (R102-A9118, R155-2014-1724, and R248-2017-2003), the EU H2020 Program (grant no. 667302; "CoCA" to A.D.B.), NIMH (1U01MH109514-01 to A.D.B.), and the Universities and University Hospitals of Aarhus and Copenhagen. The Danish National Biobank resource was supported by the Novo Nordisk Foundation. High-performance computer capacity for handling and statistical analysis of iPSYCH data on the GenomeDK HPC facility was provided by the Center for Genomics and Personalized Medicine and the Center for Integrative Sequencing, iSEQ, Aarhus University, Denmark (grant to A.D.B.). This research has been conducted using data from UK Biobank, a major biomedical database, under project 31063. This study was reviewed and approved by Partners Human Research of Partners HealthCare. The study name is Molecular Study of Cognitive and Behavioral Variation (IRB: 2015P002376), and the Principal Investigator is Elise Robinson. The authors would like to deeply thank all participants in the cohorts included in this analysis and Luke O'Connor for helpful comments.

### AUTHOR CONTRIBUTIONS

E.M.W., D.J.W., J.G., A.R., J.M.F., W.K.T., C.E.C., N.B., C.v.d.M., R.K.W., F.K.S., D.S.P., and J.B.-G. conducted data analysis, data curation, and quality control. E.M.W., D.J.W., and E.B.R. wrote the manuscript. D.M.H., P.B.M., M.J.D., M.E.T., A.D.B., and E.B.R. supervised data analysis. E.M.W., D.J.W., S.J.S., S.L.B., and E.B.R. designed the study. The members of the iPSYCH Consortium include Thomas Werge, Ole Mors, Merete Nordentoft, Thomas D. Als, and Marie Bækvad-Hansen.

### DECLARATION OF INTERESTS

D.S.P. was an employee of Genomics plc. All the analyses reported in this paper were performed as part of D.S.P.'s previous employment at the Analytic and Translational Genetics Unit, Department of Medicine, Massachusetts General Hospital, Boston, MA, USA and Stanley Center for Psychiatric Research, Broad Institute of MIT and Harvard, Cambridge, MA, USA. All other authors declare no competing interests.

### SUPPORTING CITATIONS

The following references appear in the supplemental information: Staples et al.,[59] Cann et al.,[60] Rosenberg et al.,[61] Rosenberg et al.,[62] Bergstrom et al.,[63] and O'Connell et al.,[64]

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

**Cell Genomics**
Article

CellPress

## STAR★METHODS

### KEY RESOURCES TABLE

| REAGENT or RESOURCE | SOURCE | IDENTIFIER |
|---|---|---|
| **Deposited data** | | |
| HapMap 3 | The International HapMap 3 Consortium, 2010[34] | ftp://ftp.ncbi.nlm.nih.gov/hapmap/ |
| Human Genome Diversity Project (HGDP) | Bergström et al., 2020[35] | ftp://ngs.sanger.ac.uk/production/hgdp/hgdp_wgs.20190516/ |
| SFARI-generated genotype array data | SFARI | https://www.sfari.org/resource/sfari-base/ |
| SFARI-generated whole exome sequencing data | SFARI | https://www.sfari.org/resource/sfari-base/ |
| UK Biobank genotype array data | Bycroft et al., 2018[29] | https://www.ukbiobank.ac.uk/enable-your-research/apply-for-access |
| **Software and algorithms** | | |
| ADMIXTURE | Alexander et al., 2009[36] | https://dalexander.github.io/admixture/ |
| Eagle v2.3.5 | Loh et al., 2016[37] | https://www.hsph.harvard.edu/alkes-price/software/ |
| EIGENSOFT (including smartPCA) | Price et al., 2006 Galinsky et al., 2016[38,39] | https://www.hsph.harvard.edu/alkes-price/software/ |
| Genome Analysis Toolkit (GATK) v4.1.2.0 HaplotypeCaller | GATK Team | https://hub.docker.com/r/broadinstitute/gatk/ |
| Hail | https://hail.is/ | https://github.com/hail-is/hail/ |
| IMPUTE2 | Howie et al., 2009[40] | https://mathgen.stats.ox.ac.uk/impute/impute_v2.html |
| LDpred 1.0.11 | Vilhjálmsson et al., 2015[41] | https://github.com/bvilhjal/ldpred |
| METAL | Willer et al., 2010[42] | https://genome.sph.umich.edu/wiki/METAL |
| Minimac3 | Das et al., 2016[43] | https://genome.sph.umich.edu/wiki/Minimac3 |
| picopili | Walters et al., 2018[44] | https://github.com/Nealelab/picopili |
| PLINK 1.9 | PLINK Working Group[45] | https://www.cog-genomics.org/plink/1.9/ |
| PLINK 2 | PLINK Working Group[45] | https://www.cog-genomics.org/plink/2.0/ |
| PRIMUS | Staples et al., 2013[46] | http://primus.gs.washington.edu |
| R 3.3.1 | R Core Team | https://www.r-project.org/ |
| Ricopili | Lam et al., 2020[47] | https://hub.docker.com/r/bruggerk/ricopili |
| SHAPEIT | Delaneau et al., 2011[48] | https://mathgen.stats.ox.ac.uk/genetics_software/shapeit/shapeit.html |

### RESOURCE AVAILABILITY

#### Lead contact
Further information and requests may be directed to the lead contact Elise Robinson (erob@broadinstitute.org).

#### Materials availability
This study did not generate new unique reagents.

#### Data and code availability
- The iPSYCH data reported in this study cannot be deposited in a public repository because of the sensitive nature of the data. The iPSYCH Consortium is working with GDPR compliant models for remote access. To request access, please contact authors Preben Bo Mortensen (pbm@econ.au.dk) and Anders D. Børglum (anders@biomed.au.dk) for more details.
- The imputed SPARK dataset used in this analysis has been deposited with the Simons Foundation Autism Research Initiative (SFARI) for public distribution. Scientists wishing to access the data set can do so through application to SFARI.
- Approved researchers can access UK Biobank data by applying at https://www.ukbiobank.ac.uk/enable-your-research/apply-for-access
- The HapMap 3 and HGDP data are publicly available and listed in the key resources table.
- This study did not generate original code.

**Cell Genomics**
**Article**

## EXPERIMENTAL MODEL AND SUBJECT DETAILS

### Simons simplex collection (SSC)

The SSC consists of over 2,500 simplex families with a child diagnosed with ASD.[26] We performed both family-based and case-control analyses using European ancestry individuals from SSC (see STAR Methods: Ancestry definition). For analyses without family structure (Figure 2), we analyzed 2,005 probands, 2,061 mothers and 2,079 fathers. For analyses with family structure (Figure 3), we analyzed 1,644 trios with two parents and an ASD offspring, and 1,571 trios with two parents and an unaffected sibling.

### Simons foundation powering autism research for knowledge (SPARK)

SPARK is a large-scale ongoing collection consisting of families with a child diagnosed with ASD.[27] Unlike SSC, parents in SPARK can also have an ASD diagnosis, and we subset to families where both parents do not have ASD. We performed both family-based and case-control analyses using European ancestry individuals from SPARK (see STAR Methods: Ancestry definition). For analyses without family structure (Figure 2), we analyzed 5,623 probands, 5,375 mothers, and 3,847 fathers from SPARK. For analyses with family structure (Figure 3), we analyzed 3,176 SPARK trios with two parents and an ASD offspring, and 1,559 trios with two parents and an unaffected sibling.

### UK Biobank

The UK Biobank is a cohort of 500,000 individuals living in the UK who were recruited between 2006 and 2010, aged between 40 and 69 years at recruitment. For ease of computation, we randomly selected 20,000 samples from UKB to serve as the population control cohort in our analyses.

### iPSYCH

The Danish Psychiatric Central Research Register and the Danish National Patient register, complete until 2012 and 2013, respectively, contain medical record data on the entire Danish population born between May 1, 1981 and December 31, 2005 ($n = 1,472,76$). The Lundbeck Foundation Initiative for Integrative Psychiatric Research (iPSYCH) consortium has established a large Danish population-based psychiatric case–cohort sample (iPSYCH2012) from this data to investigate the genetic and environmental architecture of severe mental disorders.[49]

## METHOD DETAILS

### Identifying families in Danish registry data

In this work, we focus specifically on ASD cases from iPSYCH ($n = 16,146$), defined as individuals with ICD-10 codes F84.0, F84.1, F84.5, F84.8 or F48.9, as well as ID cases ($n = 4,727$), defined as individuals with any ICD-10 codes from F70-F79. Controls were population representative, randomly sampled individuals from the Danish population ($n = 30,000$). Controls may have psychiatric disorders, with prevalence levels amongst controls matching those seen in the Danish general population.

The iPSYCH2012 cohort contains medical diagnoses, prescribed medicine, and social and socioeconomic data for 449,882 individuals, and their first-degree relatives. Of those, 39,491 individuals had a missing identification number for one or both of their parents or were missing phenotypic sex. In total, there were 410,391 individuals with first degree relatives for which we had phenotypic sex, and an identification number for both parents. Amongst these 410,391 individuals, we identified 274,837 families. We further subset these families to those with more than one offspring ($n = 94,790$ families).

### Sibling recurrence of ASD and ID

For each family, we selected an index case based on two criteria: (1) sex (male or female), and (2) neurodevelopmental diagnosis (*ASDnoID*, *ASDandID*, or *IDnoASD*). Families without an index case were not considered. If more than one child in a family met the given criteria, one was randomly selected as the index case, with each offspring having an equal probability of being selected as the index case.

We then selected one sibling per index case. If an index case had more than one sibling, one was randomly selected, with each sibling having an equal probability of being selected. Selected siblings were subset to those born between 1981 and 2005. Each of these siblings were matched with two age-and sex-matched Danish population representative controls. All siblings of index cases were removed from the control cohort before being matched.

We then ran logistic regressions *NDD case status* $\sim 1_{sib\ of\ case}$ (where $1_{sib\ of\ case}$ is an indicator variable for whether the individual was the sibling of an NDD case [1], or an age and sex matched control [0]), to investigate whether siblings of index cases have an increased risk for *ASDnoID*, *ASDandID*, and *IDnoASD* compared to age and sex matched controls.

ORs for increased risk with sibling case status are the exponentiated effect size for the association between sibling case status and diagnosis of a psychiatric disorder. To compare the ORs between siblings of female and male cases, we conducted a Wald test. The Wald test determines whether ORs (from the above described logistic regressions) are significantly different from one another.

This analysis was run for six types of index case: (1) female *ASDnoID*, (2) male *ASDnoID*, (3) female *ASDandID*, (4) male *ASDandID*, (5) female *IDnoASD*, and (6) male *IDnoASD*.

We performed a similar analysis to investigate increased risk of ASD diagnosis by sibling sex, selecting one ASD index case at random for each family, regardless of index case sex and comorbid ID status. If there was more than one offspring with ASD in a family, one offspring was randomly selected as the index case, with each offspring having an equal probability of being selected. Details of this analysis can be found in Methods S1: Sibling recurrence of ASD and ID, by sibling sex, Figure S1 and Table S8.

### Danish genotype data imputation

The iPSYCH2015 sample is an extension of the iPSYCH2012 sample expanding the birth cohorts by 3 years up to 2008 and extending the follow up to 2015, as well as drawing another 20,000 random samples for the random population subcohort. The new additional subsample is called iPSYCH2015i. Details of the sample, genotyping and call sets can be found in prior iPSYCH publications.[15,49,50]

Briefly, DNA was extracted from Guthrie cards in the Danish Neonatal Screening Biobank at Staten Serum Institute (SSI) and whole genome amplified. The two subsamples, iPSYCH2012 and iPSYCH2015i, were processed independently. Genotyping of the iPSYCH2012 sample was performed in 26 waves at the Broad Institute of Harvard and MIT using the PsychChip array from Illumina and the iPSYCH2015i sample was genotyped on the Global Screening Array v2 at the SSI.

Two stages of pre-imputation QC were conducted. In the first stage, we performed a near default Ricopili QC.[47] First, SNPs with a call rate < 0.95 were removed. Next, sample QC was run: we retained individuals with a call rate in cases or controls $\geq$ 0.95 and an autosomal heterozygosity deviation ($F_{HET}$) within $+/-$ 0.20 of cases or controls. Subsequently, we ran marker QC; retaining markers with call rate $\geq$ 0.98, difference in missingness $\leq$ 0.02 between cases and controls, minor allele frequency (MAF) $\geq$ 0.01, Hardy-Weinberg equilibrium (HWE) in controls (p $\geq$ 1.0 × 10$^{-6}$), and HWE in cases (p $\geq$ 1.0 × 10$^{-10}$). See https://sites.google.com/a/broadinstitute.org/ricopili/preimputation-qc for further details.

The second stage of pre-imputation QC was targeted at batch effects. In iPSYCH2012 we considered three types of potential batch effects: pre-processing plate, array plate and wave, and in iPSYCH2015i we considered pre-processing plate, array plate, and array batch. We evaluated batch effects using unrelated, ancestry matched individuals in order to avoid confounding batch effects with population stratification or cryptic relatedness. For each of the three batch types, we looped over batches, performing a GWAS of each batch against the remaining batches. Association testing was conducted using PLINK (version 1.9). The exclusion of SNPs strongly associated with any of the batch types was based on the minimum p-value across all associations per batch type. The p-value cut-off for the wave and array batch was minimum p < 2.0 × 10$^{-10}$, and for pre-processing plate and array plate, minimum p < 2.0 × 10$^{-12}$.

Imputation was performed separately for the two samples following Ricopili defaults prephasing using Eagle v2.3.5[51] and imputation using Minimac3.[43] As reference we used the public part of the Haplotype Reference Consortium[52] (EGAD00001002729) prepared for the pipeline by the Ricopili team.[47]

### Danish ASD GWAS

Our GWAS cases (n = 19,870) and controls (n = 39,078), are composed of iPSYCH2015 individuals with ASD and without ASD, respectively.

We defined sample ancestry based on a principal component analysis (PCA) using smartPCA.[38,53] We removed regions of extended linkage disequilibrium[54] (including the HLA region), and thinned the SNPs using PLINK2[45,54] by pruning those with pairwise $r^2$ > 0.075 in a window of 1000 SNPs with and step size of 100 SNPs, leaving roughly 30k markers.

Using PLINK's identity by state analysis, we identified pairs of samples with $\hat{\pi}$ > 0.2, and excluded one sample from each pair at random (with a preference for keeping cases). We restricted the cohort to individuals of European ancestry: within an ellipsoid in the space of PCs 1-3, centered on the mean of samples with all parents and grandparents born in Denmark according to national registries, and within 8 SDs along each of the first three principal axes. Following restriction to these samples, we conducted a second PCA on these individuals and used the PCs as covariates for the association analysis.

We conducted association analyses separately in iPSYCH2012 and iPSYCH2015i using PLINK on the imputed dosage data, and controlling for the first ten PCs. We meta-analyzed the results of the two ASD GWAS using METAL[42] (July 2010 version) with an inverse variance weighted fixed effect model.[55]

### SSC imputation

The imputation and QC of SSC genotype data has been described previously.[4] Each member of the family was genotyped on one of the following arrays: Illumina Omni2.5, Illumina 1Mv3, or Illumina 1Mv1 (hg19). Note that the SSC cohort only includes unaffected parents and a single ASD proband. A single unaffected sibling per family is included in analysis; if there are multiple in a family, the sibling closest in age to the proband (SSC: "designated sibling") is included.

### SPARK imputation

SPARK samples were genotyped on the Illumina Infinium Global Screening Array-24 v1.0 (GRCh38). Liftover from GRCh38 to hg19 was carried out using Hail (https://hail.is/). SPARK data were processed, restricted to individuals of European ancestry, and imputed using the Picopili pipeline[44] (https://github.com/Nealelab/picopili), which is an adaptation and extension of Ricopili[47] for family data. Phasing and imputation were conducted using SHAPEIT[48] and IMPUTE2,[40] respectively, using Haplo-type Reference Consortium[52] (HRC) data and genome build hg19. Genotypes were called for 7,124,628 autosomal SNPs (minimum posterior probability >0.8), with a genotyping rate of 0.995 across 16,965 samples of European ancestry. We removed SPARK parents with an ASD diagnosis from analysis. We

included all probands from multiplex families as well as all unaffected siblings. Additional details on genotype QC and imputation of SPARK data can be found in Methods S2: SPARK ancestry assignment, pre-imputation quality control, and imputation.

### De novo variant analysis

We downloaded gVCFs generated by GATK for 27,270 individuals from SFARIbase (/SPARK/Regeneron/SPARK_Freeze_20190912/ Variants/GATK/). All gVCFs were generated with GATK v4.1.2.0 HaplotypeCaller using default thresholds and based on hg38 reference and target files provided by Regeneron (genome.hg38rg.fa and xgen_plus_spikein.b38.bed respectively). We then performed joint calling of these 27,270 sample gVCFs via GATK to produce one unified vcf for the SPARK cohort. Subsequent variant filtering QC of SPARK data, as well as de novo variant detection, were carried out using consistent thresholds with those described previously.[7] Whole-exome sequencing and QC of SSC data has been described previously.[7,11]

We identified the ASD probands in SSC and SPARK who carried a de novo variant in a class previously associated with ASD risk.[56] These variants constitute three groups: (1) protein-truncating variants to genes intolerant of heterozygous loss of function variation (constrained gene: probability of loss of function intolerance > 0.9),[57] (2) copy number variants (deletions or duplications) affecting at least one constrained gene[4,7] and (3) predicted protein-altering missense variant in a missense constrained gene or region, defined by a Missense badness, PolyPhen-2, and Constraint (MPC) score $\geq 2$[58] (missense class B variant[4,7]). Collectively, 11.6% of SSC probands carry at least one of these variants, while 12.2% of SPARK probands carry at least one. Across SSC and SPARK, 11.2% of male probands carry at least one of these variants, while 17.4% of female probands carry at least one.

### Ancestry definition in SSC, SPARK and UKB

We randomly selected 20,000 samples from UKB to serve as the population control cohort. Using PLINK (version 1.9), we then constructed a merged file with these genotyped controls, SSC (n = 10,206), SPARK (n = 16,965) and HapMap 3[34] (n = 988) for the purpose of defining ancestry. We retained SNPs with MAF >0.01 and missingness < 0.25%. Of the remaining SNPs, we randomly sampled 10,000 for ease of computation when calculating PCs. We then used PLINK to calculate the PCs. To define ancestry, we merged all 48,159 samples, performed PCA, and selected a sub-sample of our cases and controls that clustered with Europeans in HapMap ($-0.002 <$ PC1 $< 0.003$, $-0.004 <$ PC2 $< 0.003$) (Figure S2).

We then calculated PCs in this European ancestry subset of UKB, SSC and SPARK (Figure S3). First, we retained SNPs with MAF >0.01 and missingness < 1%. Then, we performed LD pruning using PLINK to retain SNPs in approximate linkage equilibrium (–indep-pairwise 50 5 0.15). Next, we removed SNPs in 24 regions of long-range LD (mean partition size: 5.5Mb).[54] We then used PLINK to perform PCA on the remaining 95,509 SNPs and used the first 15 PCs for downstream analyses to control for ancestry.

### Generation of polygenic risk score

We used LDpred[41] (version 1.0.11) and the marginal effect sizes from the iPSYCH2015 ASD GWAS to generate a polygenic risk score, using the infinitesimal model, European ancestry subset of Hapmap 3 for LD reference, and an LD radius of 384 SNPs (per LDpred guidance). The weights from LDpred were used to calculate per sample ASD PRS using linear scoring in PLINK. There were 630,583 markers in common between the genotypes and the markers in the iPSYCH2015 ASD GWAS summary statistics, all of which were used in the polygenic risk score.

### Polygenic risk comparisons

We performed two classes of analyses to compare polygenic burden between groups. The first is a between-group comparison, where the PRS between two groups is compared using linear regression while controlling for PCs, specifically: $ASD\ PRS \sim group\ indicator + PCs\ 1 - 15$. Here, only samples of European ancestry and their PCs are used (as discussed above in "Ancestry definition"). This approach was performed for comparisons in Figure 2. The between group differences in PRS are scaled by the standard deviation of the distribution of ASD PRS in the UK Biobank controls (SD = $1.01 \times 10^{-7}$). In a similar analysis, we compared PRS between male and female cases, controlling for comorbid ID: $ASD\ PRS \sim sex + ID\ status + PCs\ 1 - 15$. The second approach is a within-family pTDT,[4] where a $t$-statistic of the deviation of the offspring's polygenic risk from the mean parent expectation is compared to the null hypothesis of 0, using a two-sided one-sample $t$-test. This approach was performed for all comparisons in Figure 3. There is no restriction of ancestry in this analysis as comparisons are within family transmission tests. Polygenic deviations are scaled by the standard deviation of the distribution of mid-parent PRS for all families with a sequenced proband in SSC + SPARK (SD = $7.25 \times 10^{-8}$). The comparison of pTDT values between groups in Figure 3 is performed as a two-sided two-sample $t$-test of each pTDT deviation distribution.

All underlying data to generate figures can be found in Tables S1–S7.

### QUANTIFICATION AND STATISTICAL ANALYSIS

The quantitative and statistical analyses are described in the relevant sections of the Method details or in the table and figure legends.

