## [Document S2. Transparent peer review records for Wigdor et al · Cell Genomics]

Title: The female protective effect against autism spectrum disorder

Author list: Emilie M. Wigdor^{1,2}, Daniel J. Weiner^{1,3}, Jakob Grove⁴⁻⁷, Jack M. Fu^{1,8}, Wesley K. Thompson⁹, Caitlin E. Carey^{1,3}, Nikolas Baya^{1,3}, Celia van der Merwe^{1,3}, Raymond K. Walters^{1,3}, F. Kyle Satterstrom^{1,3}, Duncan S. Palmer^{1,3}, Anders Rosengren^{7,10}, Jonas Bybjerg-Grauholm⁷, iPSYCH Consortium¹¹, David M. Hougaard⁷, Preben Bo Mortensen^{4,10,13,14}, Mark J. Daly^{1,3,12}, Michael E. Talkowski^{1,8}, Stephan J. Sanders¹⁵, Somer L. Bishop¹⁵, Anders D. Børglum^{4,5,10}, Elise B. Robinson^{1,3,16,17*}

1. Stanley Center for Psychiatric Research, Broad Institute of MIT and Harvard, Cambridge, Massachusetts, 02142, USA
2. Wellcome Trust Sanger Institute, Hinxton, CB10 1SA, UK
3. Analytic and Translational Genetics Unit, Department of Medicine, Massachusetts General Hospital, Boston, Massachusetts, 02114, USA
4. Center for Genomics and Personalized Medicine (CGPM), Aarhus University, Aarhus, 8000, Denmark
5. Department of Biomedicine (Human Genetics) and iSEQ center, Aarhus University, Aarhus, 8000, Denmark
6. Bioinformatics Research Centre, Aarhus University, Aarhus, 8000, Denmark
7. The Lundbeck Foundation Initiative for Integrative Psychiatric Research, iPSYCH, Aarhus, 8210, Denmark
8. Center for Genomic Medicine, Massachusetts General Hospital, Boston, Massachusetts, 02114, USA
9. Herbert Wertheim School of Public Health, University of California, San Diego, La Jolla, California, 92093, USA
10. Institute of Biological Psychiatry, MHC Sct Hans, Copenhagen University Hospital, Roskilde, 4000, Denmark
11. A list of members and affiliations appears at the end of the paper.
12. Finnish Institute for Molecular Medicine, University of Helsinki, Helsinki, 00290, Finland
13. National Center for Register-Based Research, Aarhus University, Aarhus, 8210, Denmark
14. Center for Integrated Register-based Research, Aarhus University, Aarhus, 8210, Denmark
15. Department of Psychiatry and Behavioral Sciences, UCSF Weill Institute for Neurosciences, University of California, San Francisco, San Francisco, California, 94158, USA
16. Department of Epidemiology, Harvard T.H. Chan School of Public Health, Boston, Massachusetts, 02115, USA
17. Lead contact

Summary

Initial submission: Received : July 14th, 2021

Scientific editor: Sonia Mulyil

First round of review: Number of reviewers: 2
Revision invited : Feb 1st, 2022
Revision received : Feb 26th, 2022

Second round of review: Number of reviewers: 2
Accepted : April 27th, 2022

Data freely available: Yes

Code freely available: Yes

This transparent peer review record is not systematically proofread, type-set, or edited. Special characters, formatting, and equations may fail to render properly. Standard procedural text within the editor's letters has been deleted for the sake of brevity, but all official correspondence specific to the manuscript has been preserved.

Referees' reports, first round of review

Reviewer 1:

The female protective effect against autism spectrum disorder by Wigdor et al is a straightforward manuscript examining the differences in autism by sex based upon the well-known sex difference in ASD diagnosis for individuals without intellectual disabilities. The root cause of these differences is not entirely known, and the authors use two different populations and strategies to show that females with autism have greater genetic loading, in part at least due to common variants predisposing to ASD. The study design is simple and the manuscript is short and easy to follow. The findings are intuitive and consistent with other genetic models of diseases with sex differences (the sex less frequently affected has greater genetic loading

to surpass the threshold of liability).

The authors make an important distinction between ASD +/- ID. However, the data from the Danish cohort is not particularly accurate since ID is not well documented in the medical records. Is there no way to connect educational data/standardized test scores on these same participants to get a more accurate and perhaps quantitative assessment of cognitive function?

Figure 1: Can you include the n's beneath each bar?
Why do the authors include the $P=0.12$ on the right since it's not significant?

Figure 2: It is really surprising that the female proband ASD PRS is non-significantly higher than male proband ASD PRS (0.03 SD, $p = 0.30$). What if the ASD probands were analyzed separately as those +/- ID? Can the authors analyze and compare PRS for other traits such as schizophrenia and increased educational attainment in the same subjects in figure 2?

Analyses shown in Figure 3. The definition of high impact de novos is not the traditional definition many would use. For instance CNVs (del/dup) containing at least one constrained gene is usually not the definition most would use but would focus on dosage sensitive gene since duplications of many constrained genes still may not be deleterious.

Furthermore, predicted protein-altering missense variant in a constrained gene (missense class B variant 1,4) should be more restricted regarding missense variants and should be limited to rare variants (<0.0001) that are predicted to be deleterious ($CADD>25$)

The authors briefly touch on the fact that there are likely other reasons for sex differences beyond genetics but should go into this in greater detail. There are significant biases in diagnosis based upon societal norms for behavior and the sex of evaluators, internal and externalizing features of autism that vary by sex that may not be as readily observable, and ability of non-ASD females to help ASD-females cope/compensate.

Reviewer 2:

This is a nice analysis of sex-based differences in genetic risk in autism that may explain the protective effect of female sex. The analysis used a number of large data sets to investigate common genetic risk factors that may underlie the FPE. The first observation is to show that in a population sample there is increased risk of autism to siblings associated with female autism diagnosis. The second is to show in autism datasets that mothers with autism has an increased burden of polygenic risk for autism than fathers in comparison with population based controls. A further hypothesis that there would be a difference between male and female unaffected siblings in autism PRS was not significant. The paper provides interesting analyses that will inform future investigations and the design of studies that can better address sex differences in ASD in the future.

The analyses benefit from three data sets with their own advantages and disadvantages. Analysis 1 used autistic individuals and controls derived from iPsych and the Danish birth cohort (DBC). This sample is large, representative and links genomics data and health records. Its weakness is that there are only clinical diagnosis codes available and some clinical codes are missing, eg. Diagnosis of intellectual disability which is an important variable used to define the index case in the analysis. Under ascertaining ID means that the confidence in the case definition 'ASD no ID' is reduced. This is touched on in the discussion but the implication of this for the analysis is not clearly articulated. It may be that there are overlaps between ASD no ID and ASD with ID, the latter case definition is presumably more robust. Could this explain the increased risk in siblings to ASD with ID as well as ASD no ID where the index case is defined as ASD no ID? The authors should comment on this.

Analysis 2 uses two autism datasets, SSC and SPARK. SSC benefits from excellent clinical characterisation and a quad design of parents, index case with ASD and an unaffected sibling. It is a smaller dataset which was designed to identify carriers of rare mutations and therefore polygenic risk has been observed to be lower compared with other autism datasets using different ascertained methods. It is complemented by SPARK which is larger dataset but has poorer clinical methods based on self report clinical diagnoses that are known to vary from clinically confirmed cases. The major weakness with the combined SSC/SPARK dataset is the relatively large missing data for fathers compared with mothers. As little is known about their affection status or polygenic risk, is it possible or even probable that the missing fathers could carry greater ASD polygenic risk?

A lack of datasets to replicate analysis 2 in particular is a function of dataset availability in a field that

requires very large sample sizes for studies, therefore is unavoidable.

Conceptually, they acknowledge that risk and protection cannot be disaggregated, however this might be better to acknowledge up front in the introduction as a caveat rather than waiting until the discussion.

Authors' response to the first round of review

Reviewer #1 Comment #1: The authors make an important distinction between ASD +/- ID. However, the data from the Danish cohort is not particularly accurate since ID is not well documented in the medical records. Is there no way to connect educational data/standardized test scores on these same participants to get a more accurate and perhaps quantitative assessment of cognitive function?

This is an excellent suggestion. While it would be fantastic to connect standardized tests scores to participants, unfortunately this data is not linked in the registry at present. In the main text we have added this point to the limitations in the discussion section (page 7).

Comment #2: Figure 1 - Can you include the n's beneath each bar? Why do the authors include the $P=0.12$ on the right since it's not significant? We included $P = 0.12$ in this figure to highlight the absence of a male/female sibling effect in "ID without ASD" case diagnosis category. This is in contrast to the sibling sex difference for Case diagnosis of ASD without ID. We also wanted to be consistent in our presentation of p-values.

We appreciate the reviewer's suggestion regarding the Figure 1 ns. We are concerned that adding an additional 12 ns to the Figure would make it difficult to read and interpret. Additional data regarding the specific ORs and CIs for each of the comparisons presented in Figure 1 is included in the Supplement.

Comment #3: It is really surprising that the female proband ASD PRS is non-significantly higher than male proband ASD PRS (0.03 SD, $p = 0.30$). What if the ASD probands were analyzed separately as those +/- ID?

We thank the reviewer for highlighting this potential analysis. We did not perform this analysis due to the low number of cases of ASD without ID (limited power).

Comment #4: Can the authors analyze and compare PRS for other traits such as schizophrenia and increased educational attainment in the same subjects in figure 2?

The reviewer highlights a potentially interesting set of analyses. However, the interpretability of these results is challenging given the genetic correlation between the polygenic influences on schizophrenia, educational attainment and ASD, and differences in power for the GWAS of these Response to Reviewers traits. We're developing methods to distinguish between the unique and shared components of those genetic risk scores and will present the results of that analysis in a subsequent manuscript.

Comment #5: Analyses shown in Figure 3. The definition of high impact de novos is not the traditional definition many would use. For instance CNVs (del/dup) containing at least one constrained gene is usually not the definition most would use but would focus on dosage sensitive gene since duplications of many constrained genes still may not be deleterious. Furthermore, predicted protein-altering missense variant in a constrained gene (missense class B variant 1,4) should be more restricted regarding missense variants and should be limited to rare variants (25)

We thank the reviewer for the opportunity to clarify our selection of deleterious variants in ASD. Both constrained duplications and missense class B variants are implicated in ASD liability. In detail, the recent preprint from the Autism Sequencing Consortium demonstrated a highly significant excess of duplications containing constrained genes in ASD cases vs. controls (Fu et al. 2021 medRxiv, Figure 1F). Similarly, missense class B variants are strongly implicated in ASD liability (Satterstrom et al. 2020, Cell, Figure 1B). With regard to using an allele frequency cut-off of <0.0001 for missense variants, we believe by restricting our analyses to de novo variants, we meet this requirement. Regarding restricting missense variants that are predicted to be deleterious, missense variants in class B have an MPC (for Missense badness, PolyPhen-2, and Constraint; Samocha et al., 2017, bioRxiv) score ≥ 2 , which improves variant deleteriousness prediction. Missense variants in this category are enriched in neurodevelopmental disorder cases (Samocha et al., 2017, bioRxiv, Figure 3) and MPC has shown improvement of deleteriousness prediction over CADD (Samocha et al., 2017, bioRxiv, Table 2). We have included this information in the manuscript to make explicit the deleterious effects of missense class B variants (page 13).

Comment #6: The authors briefly touch on the fact that there are likely other reasons for sex differences beyond genetics but should go into this in greater detail. There are significant biases in diagnosis based upon societal norms for behavior and the sex of evaluators, internal and externalizing features of autism that vary by sex that may not be as readily observable, and ability of non-ASD females to help ASD-females cope/compensate.

The reviewer brings up an excellent point that the reasons for sex differences in ASD are multifactorial. We have expanded this point in the discussion (page 7).

Reviewer #2

Comment #1: Analysis 1 used autistic individuals and controls derived from iPsych and the Danish birth cohort (DBC). Its weakness is that there are only clinical diagnosis codes available and some clinical codes are missing, eg. Diagnosis of intellectual disability which is an important variable used to define the index case in the analysis. Under ascertaining ID means that the confidence in the case definition 'ASD no ID' is reduced. this is touched on in the discussion but the implication of this for the analysis is not clearly articulated. It may be that there are overlaps between ASD no ID and ASD with ID, the latter case definition is presumably more robust. could this explain the increased risk in siblings to ASD with ID as well as ASD no ID where the index case is defined as ASD no ID? The authors should comment on this.

We thank the reviewer for the opportunity to clarify this point in the manuscript. In the main text we have clarified the potential quantitative impact of missing ID diagnoses (page 7). In brief, if comorbid ID were in fact present in "ASD no ID" index cases, we would expect their siblings to be more likely to receive a diagnosis, which would increase overall recurrence rates amongst siblings and bias our results towards the null.

Comment #2: The major weakness with the combined SSC/SPARK dataset is the relatively large missing data for fathers compared with mothers. As little is known about their affection status or polygenic risk, is it possible or even probable that the missing fathers could carry greater ASD polygenic risk?

Thank you for this comment, which is an opportunity to highlight an analysis that we performed to this point. On page 4, we perform the PRS comparison both for all families, as well as just for

families composed of a full trio: families where both parents are present in the dataset. We found similar results for both analyses. This analysis alleviates concerns about missingness-induced bias.

Comment #3: Conceptually, they acknowledge that risk and protection cannot be disaggregated, however this might be better to acknowledge up front in the introduction as a caveat rather than waiting until the discussion.

We thank the reviewer for the opportunity to clarify our approach. The focus of this manuscript is understanding why females are less likely to be diagnosed, hence the framing of the “female protective effect” and the framing as such in the introduction. We have modified the language in the discussion to clarify this point.

Referees' report, second round of review

Reviewer 1:

I would still like to see the analysis I suggested: What if the ASD probands were analyzed separately as those +/- ID? I think this is fundamentally an important question in autism research. I appreciate that power is limited, but it is a fundamental question about the genomic architecture of a complex diagnostic label and this is a unique cohort to address the question.

Reviewer 2:

The authors have clarified the questions I raised. I have no further comments

Authors' response to the second round of review

Reviewer1: I would still like to see the analysis I suggested: What if the ASD probands were analyzed separately as those +/- ID? I think this is fundamentally an important question in autism research. I appreciate that power is limited, but it is a fundamental question about the genomic architecture of a complex diagnostic label and this is a unique cohort to address the question.

We thank the reviewer for suggesting this analysis. We have included the results on page 4, and a description of the analysis in the STAR Methods: Polygenic risk comparisons section on page 14